# Role of Tail Dynamics on the Climbing Performance of Gecko-Inspired Robots: A Simulation and Experimental Study

**DOI:** 10.3390/biomimetics9100625

**Published:** 2024-10-14

**Authors:** Shengchang Fang, Guisong Chen, Tong Liu, Weimian Zhou, Yucheng Wang, Xiaojie Wang

**Affiliations:** 1Institute of Intelligent Machines, Hefei Institutes of Physical Science, Chinese Academy of Sciences, Hefei 230031, China; scfang@iamt.ac.cn (S.F.); chengs@iim.ac.cn (G.C.); lt1997@mail.ustc.edu.cn (T.L.); wmzhou@mail.ustc.edu.cn (W.Z.); 2University of Science and Technology of China, Hefei 230026, China

**Keywords:** gecko-inspired robots, tail dynamics, dynamic simulation, climbing performance

## Abstract

Geckos are renowned for their exceptional climbing abilities, enabled by their specialized feet with hairy toes that attach to surfaces using van der Waals forces. Inspired by these capabilities, various gecko-like robots have been developed for high-risk applications, such as search and rescue. While most research has focused on adhesion mechanisms, the gecko’s tail also plays a critical role in maintaining balance and stability. In this study, we systematically explore the impact of tail dynamics on the climbing performance of gecko-inspired robots through both simulation and experimental analysis. We developed a dynamic climbing simulation system that models the robot’s specialized attachment devices and predicts contact failures. Additionally, an adjustable-angle force measurement platform was constructed to validate the simulation results. Our findings reveal the significant influence of the tail on the robot’s balance, stability, and maneuverability, providing insights for further optimizing climbing robot performance.

## 1. Introduction

Geckos are natural climbing experts, capable of agile movement across various rugged environments. This ability is primarily due to their specialized feet, which feature hairy toes that can uncurl to attach to surfaces using van der Waals forces and peel off within milliseconds [1,2]. As a result, geckos have become a significant source of inspiration for the development of climbing robots, such as Geckobot [3] and Stickybot [4,5]. These bio-inspired robots have demonstrated immense potential and importance in a range of applications, particularly in high-risk tasks such as search and rescue, inspection and maintenance, and military reconnaissance.

Currently, most research has focused on developing adhesion mechanisms, including suction cups [6,7,8], magnets [9,10,11], micro-spines [12,13], and dry adhesive materials [14,15]. Based on these mechanisms, many climbing robots have demonstrated reliable climbing capabilities. However, the gecko’s remarkable climbing ability is not solely due to the setae on its toe pads [16]; its tail also plays a crucial role. For instance, when a gecko slips during rapid climbing, it quickly presses its tail against the vertical surface to prevent its head from pitching backward. Even if the backward pitch cannot be fully prevented, the gecko positions its tail like a bicycle kickstand to avoid falling [17]. Additionally, some geckos use their tails to adjust their direction while gliding, allowing them to navigate through trees [18]. Geckos can also increase their speed by adjusting the amplitude and frequency of lateral undulations in their bodies and tails [19,20]. These biological phenomena suggest that the tail has significant potential for further enhancing the locomotion performance of climbing robots.

In this paper, our aim is to systematically explore the impact of the tail on the climbing performance of robots through experimental research. However, conducting experiments on climbing robots designed for high-risk tasks is often challenging, as it typically requires large-scale equipment to ensure the safety of both the robot prototypes and the operators. To address this, we developed a climbing dynamics simulation system designed to replicate the robot’s real climbing behavior in a simulated environment. Unlike existing dynamic climbing models that simplify the interaction between the robot’s feet and the wall as stable contact [21,22,23], the core design of our simulation system focuses on dynamic contact modeling of the robot’s specialized attachment devices. The model effectively predicts contact failures during the climbing process and outputs key parameters such as the robot’s motion trajectory, contact forces, and joint torques.

Additionally, we developed an adjustable-angle force measurement platform to test the robot’s climbing performance in real-world environments, further validating the effectiveness of the simulation model. By comparing the robot’s performance with and without a tail, this paper explores the impact of the tail on the balance, stability, and maneuverability of climbing robots. The insights gained from the simulations and experimental results, along with the subsequent analysis, will contribute to the further optimization of the robot’s climbing performance.

## 2. Construction of a Climbing Dynamics Simulation System

In our previous work, we designed a gecko-inspired climbing robot, WCQR-III [24], by mimicking the body structure of a gecko. Based on parameters such as the robot’s dimensions, weight, and attachment mechanism, we constructed a climbing dynamics simulation system. This system was used to simulate the robot’s climbing behavior in a virtual environment and to predict potential climbing failures in both tailed and tailless states.

### 2.1. Overall Structure of the WCQR-III Robot

Figure 1a shows the overall structure of a gecko, with its body connected to four legs and a tail. Each leg consists of three main parts: the thigh, crus, and claw, which are connected by various joints. The thigh is connected to the body via a hip joint, allowing for two degrees of freedom in spherical motion. The thigh and crus are connected by a knee joint, which provides one degree of freedom in rotational motion. The claw at the end is used for climbing surfaces. The tail is connected to the body through a tail joint. By abstracting the gecko’s configuration, we designed the climbing robot WCQR-III, as shown in Figure 1b. The robot features four legs and a tail, with a flat body. The body is equipped with a circuit board, and the onboard battery is mounted at the bottom of the body to keep the center of gravity as close to the wall as possible, minimizing pitch moments caused by shifts in the center of gravity. Each leg of the robot has three active joints, each driven by an actuator (Model HV0916, Yobotics Co., Ltd., Jinan, China) with a rated output torque of 6.07 N·m. Unlike the gecko’s hip joint, which has two degrees of freedom, we designed a combination of a lateral hip joint and a forward hip joint to achieve multi-degree-of-freedom control. The lateral hip joint controls the up-and-down movement of the crus, while the forward hip joint controls the forward and backward movement of the crus. The robot’s tail is connected to the body via a torsion spring, and a roller is attached to the end of the tail to reduce friction with the wall. The tail can be manually removed to simulate the condition of “tail loss”. The robot’s body components are made of lightweight carbon fiber and nylon materials, keeping the total weight of the robot under 3 kg. Table 1 provides the parameters of key components.

In addition to the setae structures used for climbing on smooth surfaces, geckos possess sharp claws that provide adhesion on rough surfaces. To mimic this, we designed spiny claws as the robot’s attachment devices, enabling it to climb rough surfaces, as shown in Figure 1c. An under-actuated compliance ankle connects the spiny toepad array to the crus, allowing the spiny claw to easily contact wall surfaces. Each spiny claw consists of an array of 50 spiny toepads, with each spiny toepad design adapted from reference [25], inspired primarily by the spines on insect legs (as shown in Figure 1d). The steel hooks at the top can engage with microscopic particles on rough surfaces, forming a mechanical interlock that generates adhesion. The rigid linkage is made of nylon and is connected using flexible polyurethane material. This hybrid configuration of rigid and flexible components effectively enhances adhesion stability.

After obtaining the overall structural parameters of the robot and the adhesion mechanism of the attachment devices, we will develop a multibody dynamics model of the robot’s body structure and a dynamic contact model of the attachment devices. Finally, we will integrate these models to construct a comprehensive climbing dynamics simulation system.

### 2.2. Multibody Dynamics Modeling

Based on the parameters provided in Table 1, the overall structure of the robot can be simplified into links of different lengths with masses, connected by rotational joints, as shown in Figure 2. Considering that the robot is a complex multi-joint system, we use Kane’s method for multibody dynamics modeling. Unlike the Newton-Euler and Lagrange’s methods, Kane’s method does not calculate constraint forces and energy, significantly reducing the computational load of the dynamics [26]. This makes it particularly suitable for systems with multiple degrees of freedom.

For a complete system composed of *n* particles with *m* generalized degrees of freedom, the twist of the *j*-th degree of freedom in the body frame $\left[L_{ij}\right]$ of the *i*-th particle is:(1)$$u_{ij}=\left[\begin{array}{c} q_{ij}\times \omega_{ij}\\ \omega_{ij} \end{array}\right],\;\;i=1,2,\ldots ,n;j=1,2,\ldots ,m$$
where $\omega_{ij}$ and $q_{ij}$ respectively represent the unit vector in the positive direction and the generalized coordinate vector of the *j*-th degree of freedom in the body frame $\left[L_{ij}\right]$ of the *i*th particle. The generalized active force $F_{j}$ and the generalized inertia force $F_{j}^{*}$ of the system on the *j*-th degree of freedom is
(2)$$\left\{\begin{array}{c} F_{j}=\sum_{i=1}^{n}K_{i}\cdot u_{ij},\\ F_{j}^{*}=\sum_{i=1}^{n}K_{i}^{*}\cdot u_{ij}, \end{array}\right.\;i=1,2,\ldots ,n;j=1,2,\ldots ,m$$

Thus, we can obtain the Kane equation of the *j*-th degree of freedom of the system:(3)$$F_{j}+F_{j}^{*}=0.$$

Next, we calculate the active wrenches for each link. Based on the different force conditions, the robot’s links can be categorized into the body, links not in contact with the wall, and links in contact with the wall. First, we calculate the active wrenches $K_{ij}$ for the links not in contact with the wall:(4)$$K_{ij}=\left(\begin{array}{c} M_{ij}\\ R_{ij} \end{array}\right)=\left(\begin{array}{c} \tau_{ij}\omega_{ij}-\tau_{i(j+1)}\omega_{i(j+1)}\\ m_{ij}g \end{array}\right),\;i=1,2,3,4;j=1,2.$$
where $\tau_{ij}$ is the *j*-th joint torque of the *i*-th leg, $\omega_{ij}$ is the unit vector in the positive direction of the *j*-th joint of the *i*-th leg, and $m_{ij}$ is the mass of the *j*-th link of the *i*-th leg.

Considering the contact forces between the foot and the wall, the active wrenches $K_{i3}$ for the links in contact with the wall are given by:(5)$$K_{i3}=\left(\begin{array}{c} M_{i3}\\ R_{i3} \end{array}\right)=\left(\begin{array}{c} \tau_{i3}\omega_{i3}+F_{fi}\times \left(r_{fi}-r_{ci}\right)\\ m_{i3}g+F_{fi} \end{array}\right),\;i=1,2,3,4.$$
where $F_{fi}$ is the contact force between the foot on the *i*-th leg and the wall, $r_{fi}$ is the position vector of the end-link on the *i*-th leg, and $r_{ci}$ is the position vector of the center of mass of the end-link on the *i*-th leg.

Similarly, the active wrench $K_{b}$ for the robot’s body is calculated as follows:(6)$$K_{b}=\left(\begin{array}{c} M_{b}\\ R_{b} \end{array}\right)=\left(\begin{array}{c} -\sum_{i=1}^{4}\tau_{i1}\omega_{i1}-\tau_{t}\omega_{t}\\ m_{b}g \end{array}\right)$$
where $\tau_{t}=k_{t}\alpha$, $k_{t}$ is the elastic coefficient of the tail joint torsion spring, α is the rotational angle of the tail, and $\omega_{t}$ is the unit directional vector of the tail joint.

In order to obtain the inertial wrench, the required angular velocity and the angular acceleration of each link via the following recursive formulas are given as:(7)$$\left\{\begin{array}{c} \omega_{L_{ij}}^{b}=R_{L_{ij}L_{i(j-1)}}\cdot \omega_{L_{ij-1}}^{b}+{\dot{\theta }}_{ij}\cdot \omega_{ij}\\ {\dot{\omega }}_{L_{ij}}^{b}=R_{L_{ij}L_{i(j-1)}}\cdot {\dot{\omega }}_{L_{i(j-1)}}^{b}+{\ddot{\theta }}_{ij}\cdot \omega_{L_{ij}}+{\dot{\theta }}_{ij}\cdot \left(\omega_{L_{i(j-1)}}^{b}\times \omega_{L_{ij}}\right)\\ {\dot{v}}_{L_{ij}}^{b}=R_{L_{ij}L_{i(j-1)}}\left\{{\dot{v}}_{L_{i(j-1)}}^{b}+{\dot{\omega }}_{L_{i(j-1)}}^{b}\times \left(-l_{i(j-1)}\right)+\omega_{L_{i(j-1)}}^{b}\times \left[\omega_{L_{i(j-1)}}^{b}\times \left(-l_{i(j-1)}\right)\right]\right\}\\ {\dot{v}}_{C_{ij}}^{b}={\dot{v}}_{L_{ij}}^{b}+{\dot{\omega }}_{L_{ij}}^{b}\times l_{ij}^{b}+\omega_{L_{ij}}^{b}\times (\omega_{L_{ij}}^{b}\times l_{ij}^{b}) \end{array}\right.,\;i=1,2,3,4;j=1,2,3.$$
where $\omega_{L_{ij}}^{b}$ and ${\dot{\omega }}_{L_{ij}}^{b}$ are the angular velocity and the angular acceleration of the *j*-th link on the *i*-th leg in the body frame $\left[L_{ij}\right]$, ${\dot{v}}_{L_{ij}}^{b}$ is the acceleration in the body frame $\left[L_{ij}\right]$, ${\dot{v}}_{C_{ij}}^{b}$ is the acceleration of the CM in the body frame $\left[L_{ij}\right]$, $R_{L_{ij}L_{ij-1}}$ is the rotational transferring matrix of the frame $\left[L_{ij}\right]$ relative to the frame $\left[L_{i(j-1)}\right]$, and $l_{ij}^{b}$ is the distance from the CM to the joint of the j-th link on the i-th leg.

The inertial wrench of the body can be expressed as:(8)$$K_{b}^{*}=\left(\begin{array}{c} M_{b}^{*}\\ R_{b}^{*} \end{array}\right)=\left(\begin{array}{c} -I_{b}{\dot{\omega }}_{b}-\omega_{b}\times I_{b}\cdot \omega_{b}\\ -m_{b}{\dot{v}}_{b} \end{array}\right)=\left(\begin{array}{cc} -I_{b} & 0\\ 0 & -m_{b} \end{array}\right){\ddot{q}}_{b}+\left(\begin{array}{c} -\omega_{b}\times I_{b}\cdot \omega_{b}\\ 0 \end{array}\right)$$

The inertial wrench of the links can be expressed as:(9)$$K_{ij}^{*}=\left(\begin{array}{c} M_{ij}^{*}\\ R_{ij}^{*} \end{array}\right)=\left(\begin{array}{c} -I_{ij}{\dot{\omega }}_{L_{ij}}-\omega_{L_{ij}}\times I_{ij}\cdot \omega_{L_{ij}}\\ -m_{ij}{\dot{v}}_{C_{ij}}^{b} \end{array}\right)=\left(\begin{array}{cc} -I_{ij} & 0\\ 0 & -m_{ij} \end{array}\right){\ddot{q}}_{ij}+\left(\begin{array}{c} -\omega_{L_{ij}}\times I_{ij}\cdot \omega_{L_{ij}}\\ 0 \end{array}\right)$$
where $I_{ij}$ is the inertia matrix of link *j* on the *i*-th leg with respect to the body frame $\left[L_{ij}\right]$, $I_{b}$ is the inertia matrix of the robot body in the global frame [*O*_g_], and ${\dot{\omega }}_{b}$$\omega_{b}$${\dot{v}}_{b}$ represents the angular acceleration, angular velocity, and linear acceleration of the robot body in the global frame [*O*_g_], respectively.

According to Equations (1)~(3), the Kane equations for the robot’s body and the *i*-th leg can be obtained. By combining these equations, the complete set of dynamic equations for the robot is established:(10)$$\left\{\begin{array}{c} \sum_{i=1}^{6}(F_{bi}+F_{bi}^{*})=\sum_{i=1}^{6}\left(\left[\begin{array}{c} M_{b}\\ R_{b} \end{array}\right]\cdot u_{bi}+\left[\begin{array}{c} M_{b}^{*}\\ R_{b}^{*} \end{array}\right]\cdot u_{bi}\right)=0,\\ \sum_{i=1}^{3}(F_{ni}+F_{ni}^{*})=\sum_{i=1}^{3}\left(\sum_{j=1}^{3}\left[\begin{array}{c} M_{ni}\\ R_{ni} \end{array}\right]\cdot u_{ij}+\sum_{j=1}^{3}\left[\begin{array}{c} M_{ni}^{*}\\ R_{ni}^{*} \end{array}\right]\cdot u_{ij}\right)=0,(n=1,2,3,4) \end{array}\right.$$

### 2.3. Dynamic Contact Modeling

The Contact modeling is a critical aspect of dynamic analysis, especially in climbing dynamics, where the unique adhesion mechanisms of the foot attachment devices must be considered. For the spiny claw used by the WCQR-III, it can be simplified into a hybrid spring-damper linkage model, as shown in Figure 3b. This model represents the moment when the claw first makes contact with the wall. The model consists of two links, labeled as Link ① and Link ②, which are connected by a hinge joint. Link ① is directly connected to the robot’s foot, and the end of Link ② is equipped with a steel hook designed to make contact with the wall. A torsion spring, with stiffness $k_{\beta }$ and damping coefficient $b_{\beta }$, is installed between the two links. Additionally, a tension spring with stiffness $k_{a}$ and damping coefficient $b_{a}$ is used to allow for the extension and retraction of Link ②. In the initial state, Link ② has a length of $a_{0}$ and forms an initial angle $\beta_{0}$ with the *y*-axis. Table 2 presents the specific parameters of the model.

During the robot’s climbing process, the dynamic contact behavior of the spring-damper linkage model follows the foot trajectory shown by the blue dashed line in Figure 3c, progressing through three stages: (a) pressing, (b) dragging, and (c) detachment. In the pressing stage, the foot moves along the blue dashed line in the −z direction, starting from the initial point $C_{0}\left(0,z_{0}\right)$. When the foot reaches point $C_{1}\left(0,z_{1}\right)$, the steel hook at the tip of the spiny toepad first makes contact with the wall. The foot is then further pressed against the wall to ensure full contact between the spiny toepad and the wall surface. Once sufficient contact is achieved, the foot, under load, will move downward along the −y direction, entering the dragging phase. The steel hook at the end will mechanically interlock with the micro-particles on the rough wall surface, generating a contact force $\left(F_{a},F_{\beta }\right)$. The components of this contact force along the *y*-axis and *z*-axis are denoted as $\left(F_{x},F_{y}\right)$. Under the load, the tension spring stretches, causing the length of Link ② to change to a(t), with an elongation of y(t) in the *y*-axis direction. The amount of elongation is determined by the load applied to the spiny toepad. However, if the load is too large, the tension spring may break. Therefore, we set a maximum elongation limit for the tension spring at 10 mm. Specifically, if $y\left(t\right)<-10\;\mathrm{mm}$, the spring will be overstretched, leading to breakage and resulting in attachment failure. Due to the directional nature of the spiny toepad, it does not generate adhesion when moving in the direction opposite to the applied load [25]. Therefore, during the detachment phase, as the toepad moves upward in the reverse direction to $y\left(t\right)>0$, the link returns to its initial length, completing the unloading process, and the contact force drops to zero.

By analyzing the dynamic contact process of the spiny toepad, the scenarios in which the contact force becomes zero at different stages can be inferred based on the relative position of the toepad to the wall. The following equations describe these conditions:(11)$$F=\left\{\begin{array}{c} 0\\ 0\\ non-zero \end{array}\begin{array}{c} if\;z(t)>z_{1}\\ if\;y(t)>0\;\;\;\;or\;\;\;y(t)<-10\;\;\;and\;\;z(t)\leq \\ if\;\;-10\leq y(t)\leq 0\;\;and\;\;z(t)\leq z_{1}\; \end{array}\right.z_{1}$$

Excluding cases where the contact force is zero, we calculate the changes in the foot’s contact force at different moments when the spiny toepad forms effective adhesion. Subsequently, we establish the corresponding dynamic contact mathematical model.

Figure 3d illustrates the deformation of the spiny toepad after stable adhesion has been achieved. After undergoing the pressing and dragging actions, the length of Link ② and its angle with the wall can be expressed by the following equations:(12)$$a(t)=\sqrt{z{\left(t\right)}^{2}+{\left(y_{1}+y(t)\right)}^{2}}$$
(13)$$\beta (t)=\arctan\frac{z(t)}{y_{1}+y(t)}$$

Furthermore, the elongation speed of Link ② and the angular velocity of its angle can be expressed as follows:(14)$$\dot{a}(t)=\frac{z(t)\cdot \dot{z}(t)+(y_{1}+y(t))\cdot \dot{y}(t)}{\sqrt{z{\left(t\right)}^{2}+{\left(y_{1}+y(t)\right)}^{2}}}$$
(15)$$\dot{\beta }(t)=\frac{\dot{z}(t)\cdot (y_{1}+y(t))-z(t)\cdot \dot{y}(t)}{{\left(y_{1}+y(t)\right)}^{2}+z{\left(t\right)}^{2}}$$

The forces exerted by the steel hook on the rough wall surface are:(16)$$F_{a}=k_{a}\cdot (a(t)-a_{0})+\dot{a}(t)\cdot b_{a}$$
(17)$$F_{\beta }=\frac{k_{\beta }\cdot (\beta (t)-\beta_{0})+\dot{\beta }(t)\cdot b_{\beta }}{a(t)}$$

After converting the calculation results to the y-z coordinate system, the tangential force $F_{y}$ and the normal force $F_{z}$ on the spiny toepads can be expressed as:(18)$$\left[\begin{array}{c} F_{y}\\ F_{z} \end{array}\right]=\left[\begin{array}{cc} -\sin\beta (t) & -\frac{1}{a(t)}\cos\beta (t)\\ \cos\beta (t) & -\frac{1}{a(t)}\sin\beta (t) \end{array}\right]\cdot \left(\left[\begin{array}{c} k_{a}\cdot (a(t)-a_{0})\\ k_{\beta }\cdot (\beta (t)-\beta_{0}) \end{array}\right]-\left[\begin{array}{c} b_{a}\cdot \dot{a}(t)\\ b_{\beta }\cdot \dot{\beta }(t) \end{array}\right]\right).$$

The lateral force of the spine in the x-direction can be calculated based on static equilibrium. Figure 3c illustrates the force situation of the spine in the x-y plane. $F_{G}$ represents the load on the spiny toepad. The contact force of the spiny toepad in the *x*-axis direction is:(19)$$F_{x}=\mu F_{y}$$
where μ is the friction coefficient between the metal and the rough rock.

Considering the adhesion probability of a spiny claw composed of n spiny toepads, the dynamic contact model equation for the spiny toepad can be expressed as:(20)$$F_{f}=n\cdot P\cdot {\left[F_{x},F_{y},F_{z}\right]}^{T}$$

Its inputs are the displacement and velocity of the spiny toepad relative to the wall, while its outputs are the adhesion forces generated by the spiny toepad in the tangential, normal, and lateral directions.

### 2.4. Simulation Environment Setup

To visually demonstrate the robot’s climbing process, a simulation environment was created using the multibody dynamics software RecurDyn (Student Version, FunctionBay), which can generate simulation animations. In this environment, contact constraints between the robot’s body and the wall were established. The developed multibody dynamics and dynamic contact equations were input into MATLAB for mathematical computation. A co-simulation with RecurDyn was then conducted to calculate the changes in the robot’s motion parameters during the climbing process.

The robot’s climbing dynamics algorithm is shown as Algorithm 1. First, the desired motion trajectory is input, and the joint angle parameters of the robot are solved using inverse kinematics. Then, the center of mass trajectory of the robot is calculated using the dynamics Equation (10). After determining the current position of the foot through coordinate transformation, Equation (11) is used to assess whether effective contact has occurred. If effective contact is confirmed, the dynamic contact model is used to calculate the contact forces at the foot. Finally, the robot’s foot contact forces, joint torques, and center of mass coordinates are output.

**Algorithm 1** Dynamic Climbing Algorithm**Input:** desired motion trajectory coordinates
**Output**

$\text{:}\;\mathrm{contact}\;\mathrm{forces}\;F_{f}$


$;\;\mathrm{joint}\;\mathrm{torques}\;\tau ;\;\mathrm{CM}\;\mathrm{coordinates}\;x_{c}\left(t\right),y_{c}\left(t\right),z_{c}(t)$

1  #Inverse Kinematics (*IK*) calculation to obtain joint angles2  **for** t = 0; t ≤ T; t++ **do**3     
$\theta_{ij}\leftarrow IK(x\left(t\right),y\left(t\right),z(t))$
4     **if** −10 ≤ y(t) ≤ 0 and z(t) ≤ $z_{1}$5     
$\;\mathrm{CM}\;\mathrm{coordinates}\;x_{c}\left(t\right),y_{c}\left(t\right),z_{c}\left(t\right)\leftarrow use\;$Equation (10)
6     
 Joint torques τ←use Equation (10)
7     
$\;\mathrm{Contact}\;\mathrm{forces}\;F_{f}\leftarrow use\;$Equation (20)
8      
**break;**
9     **else**:10     
$\;\mathrm{Contact}\;\mathrm{forces}\;F_{f}\leftarrow 0$
11  
**continue**


After constructing the simulation environment, the robot’s climbing motion within the simulated environment can be observed by inputting the desired motion trajectory. We modeled the robot’s desired motion trajectory after the movement pattern of a gecko. For quadrupedal climbing animals like geckos, the primary gait used is the diagonal gait, where two diagonal legs swing simultaneously in an alternating pattern [20], as shown in Figure 4a. Unlike the rapid alternating gait used for running on flat ground, during the climbing process, a gecko first ensures that the two diagonal legs in the air securely attach to the surface before detaching the legs that were previously in contact with the wall. This approach ensures that at least two legs remain in contact with the wall at all times, providing stable adhesion throughout the climb.

Similarly, we adopted this gait pattern for the robot and divided the foot movement into four phases: attachment, stance, detachment, and swing. To ensure that the robot always maintains contact with the wall with at least two legs, we introduced a Pause phase, as illustrated in Figure 4b,c. Figure 4d shows the predefined foot trajectory. We set the robot’s gait cycle to 4 s. The horizontal displacement (x) is set to zero to ensure that the robot climbs straight up; the vertical displacement (y), representing the step length, is set to 35 mm; and the normal displacement (z), representing the step height, is set to 15 mm.

## 3. Simulation Results

To assess the impact of the tail on climbing performance, we tested the robot’s motion parameters under both tailed and tailless conditions at different inclines. The previously defined foot trajectory parameters were input into the developed climbing dynamics simulation system, with the simulation time set to 16 s to simulate the robot’s climbing behavior over four cycles. The slope variations were simulated by modifying the direction of the gravitational acceleration parameter ***g***.

### 3.1. Maximum Climbing Angle Simulation

Figure 5 presents the time sequence diagrams of the robot climbing in a simulated vertical environment under both tailed and tailless conditions. In the tailed condition, the robot successfully climbs vertically along the predefined trajectory without falling. However, in the tailless condition, the robot begins to pitch backward after 2 s of upward movement, causing the front claws to gradually detach from the wall. Due to the constraints of the wall, the rear of the robot’s body tilts to the right side after colliding with the wall, ultimately leading to a fall at the 3-s mark, preventing further upward climbing.

The robot is unable to climb vertical surfaces without its tail. To determine the maximum incline the robot can overcome in a tailless state, we gradually reduced the slope angle, starting from 90°, and simulated the climbing process at each angle. Only when the slope was reduced to 86° was the robot able to stably climb for four gait cycles without its tail, as shown in Figure 6.

Figure 7 shows the variation in contact forces at each foot of the robot after four gait cycles of climbing on a 90° vertical surface with the tail. It also presents the contact force variation at each foot while climbing an 86° slope without the tail. Solid lines represent the contact forces in the tailed condition, while dashed lines represent the contact forces in the tailless condition. Through comparison, it can be observed that the robot generates more stable adhesion forces in the tailed condition. In contrast, without the tail, the contact forces at the feet exhibit noticeable fluctuations, and the magnitudes of these forces in all directions are greater than those in the tailed condition. Further comparison of the forces in each direction reveals the following:

① For the lateral forces $F_{x}$, the left and right claws of the robot generate equal but opposite lateral forces, indicating that the limbs tend to converge toward the body’s centerline during the climbing process. Additionally, the front claws generate greater forces than the rear claws. In the tailed condition, the lateral forces produced by the front and rear claws are 4.3 N and 3.2 N, respectively. In the tailless condition, the corresponding lateral forces increase to 5.1 N and 4.2 N.

② For the tangential forces $F_{y}$, regardless of whether the robot has a tail, the front claws generate greater contact forces than the rear claws. In the tailed condition, the tangential forces produced by the front and rear claws are 17.3 N and 12.8 N, respectively. In the tailless condition, these forces increase to 20.4 N for the front claws and 12.5 N for the rear claws.

③ For the normal forces $F_{z}$, the front claws generate greater adhesion forces than the rear claws. In the tailed condition, the robot’s front and rear claws produce negative normal forces of −2.9 N and −2.3 N, respectively, indicating adhesion forces directed toward the wall. In the tailless condition, the robot requires greater normal adhesion forces to achieve stable climbing, with the front and rear claws generating contact forces of −3.24 N and −2.5 N, respectively.

### 3.2. Performance on Different Slopes

Figure 8 shows the changes in the center of mass (CoM) trajectory of the robot as it climbs slopes of 0°, 30°, 60°, and 90° in the tailed condition. For the lateral direction (*x*-direction), although the predefined foot trajectory has no lateral displacement, Figure 8a reveals that the robot’s lateral deviation increases as the climbing angle increases. Notably, during vertical climbing at 90°, the robot exhibits periodic lateral oscillations of approximately 2 mm with each upward step.

Figure 8b shows the variation in tangential displacement (*y*-direction) over time as the robot climbs different slopes. Throughout the entire motion, the robot takes a total of seven steps, with the displacement distance gradually decreasing as the slope increases. On a 0° surface, the robot moves a total distance of 244 mm, with an average step length of 34.8 mm. However, when the slope reaches 90°, the robot covers a distance of 196 mm, with an average step length of 28.0 mm, which is less than the planned step length of 35 mm. This reduction is due to increased slippage with each step as the slope increases.

Figure 8c shows the displacement of the robot in the normal direction (*z*-axis), where the robot’s center of mass exhibits periodic oscillations during movement. As the slope increases, the amplitude of these oscillations also increases. When climbing a 90° surface, the oscillation amplitude reaches approximately 1 mm. Figure 8d presents the robot’s movement trajectory in the x-y plane.

Figure 9 shows the changes in the center of mass (CoM) trajectory of the robot as it climbs slopes of 0°, 30°, 60°, and 86° in the tailless condition. As shown in Figure 9a, compared to the tailed condition, the robot exhibits greater lateral oscillations in the tailless condition. When the slope reaches 86°, the maximum lateral amplitude is approximately 6 mm. Conversely, in the tangential direction, the robot’s tangential displacement in the tailless condition is smaller than in the tailed condition, as shown in Figure 9b. On a 0° slope, the total displacement over 16 s is 242 mm, while on an 86° slope, it is 179 mm, corresponding to average step lengths of 34.6 mm and 25.6 mm, respectively. This indicates that the removal of the tail increases slippage with each step. Figure 9c shows the displacement in the normal direction under the tailless condition. Across different slopes, the robot’s normal displacement remains consistent, exhibiting a normal oscillation of approximately 1.5 mm. The reason for this consistent normal trajectory is that the robot’s rear leg foothold is positioned in the middle of the body rather than at the rear. Without a tail, the robot is unable to maintain balance in terms of pitch angle, leading to the maximum normal displacement during climbing on any slope. Figure 9d shows the robot’s movement trajectory in the x-y plane. Compared to the tailed condition, the robot experiences greater displacement and lower stability during climbing in the tailless condition.

## 4. Experiments

In addition to the analysis conducted in the simulation environment, we further developed an experimental platform to test the physical prototype in a real-world setting. This allowed us to analyze the changes in the robot’s performance with and without a tail under actual operating conditions.

### 4.1. Experimental Platform

During movement, the ground reaction force (GRF) at the feet can reflect the dynamic characteristics of animals or robots [27,28]. To analyze the impact of tail removal on the robot’s dynamic performance, we measured the GRF under both tailed and tailless conditions at different angles. Given the robot’s large geometric size, a single force sensor is insufficient to fully capture the contact forces on each leg. Therefore, we employed an array of multiple force sensors to measure the contact forces at the feet during climbing. The overall structure of the measurement platform is shown in Figure 10.

The platform’s rough surface is composed of 3 × 4 metal plates coated with rough sand particles, with each plate mounted on a three-dimensional force sensor to measure the contact forces as the robot climbs on the rough surface. The parameters of these sensors are listed in Table 3. The measurement platform has a total length of 1200 mm and a width of 900 mm. The angle can be adjusted from 0° to 180° using a drive motor. A control cabinet is used to control the motor’s rotation, while a computer connected to the sensor acquisition system reads and records the measurement data.

The WCQR-III physical prototype uses an open-loop control system and is equipped with a remote-control system. An STM32h7 chip was selected as the microcontroller. The servomotors on each joint of the robot are connected via the Controller Area Network (CAN) communication serial port. A wireless transmission module was designed for 4G mobile communication based on TDD-LTE, and the main board communicates with the remote controller through the RS232 protocol. The operator can remotely control the robot’s direction and speed.

### 4.2. Experimental Results

#### 4.2.1. Maximum Climbing Angle Test

To assess the impact of the tail on the robot’s vertical climbing ability, we tested whether the physical prototype could achieve stable climbing on a 90° vertical surface under both tailed and tailless conditions. The robot prototype’s motion parameters were based on the desired trajectory shown in Figure 4d, with a gait cycle of 4 s and a step length of 35 mm, matching the input trajectory used in the simulation.

Figure 11a shows the robot’s climbing process on a 90°vertical rough surface with a tail. The results indicate that the robot stably climbed 178 mm in 16 s, which is very close to the simulation value of 196 mm. Figure 11b presents the climbing results without the tail. After removing the robot’s tail, we used the same motion parameters to control the robot’s climbing on the vertical wall. The results show that after 3 s of climbing, the robot’s body began to pitch backward, and the front claws gradually detached from the wall. Subsequently, at the 4-s mark, the left front claw completely lost contact with the wall, failing to provide adhesion, which caused the robot to tilt to the left and fall from the wall. The experimental results are consistent with the simulation: without the tail, the robot is unable to achieve stable climbing on a 90° vertical surface.

To determine the maximum slope the robot can climb without a tail, we gradually reduced the platform’s incline angle and tested the robot’s ability to climb stably at different slopes. For each degree of reduction, three tests were conducted, and the slope angle at which the robot did not fall or exhibit significant slippage in all three tests was considered the maximum climbable slope. The results, shown in Figure 11c, indicate that the robot was able to climb stably without a tail at an incline of 82°. Within 16 s, the robot successfully moved 174 mm along the surface.

#### 4.2.2. Motion Performance Testing on Different Slopes

As shown in Figure 12, we set the platform’s incline angles to 0°, 30°, 60°, and 90°, and measured the robot’s ground reaction forces (GRF) during climbing. Since the robot is unable to climb a 90° vertical surface without a tail, the corresponding platform angles for the tailless condition were set to 0°, 30°, 60°, and 82°. Considering the potential uneven distribution of surface roughness particles, which could affect the robot’s foot forces during each climb, we conducted five climbing tests at each angle and collected the measurement data.

Since the robot uses a diagonal gait, the contact forces on the left and right feet are the same in magnitude but offset by one cycle. After the robot achieved stable attachment, we measured the GRFs at different angles, conducting five measurements in total. All data are presented as average values with error bands. Figure 13 shows the ground reaction forces (GRF) at the robot’s feet while climbing at different angles, both with and without a tail. The dashed lines represent the simulation values, while the solid lines represent the actual measured values.

For the lateral forces $F_{x}$, the simulation results show that as the incline angle increases, the lateral forces at the claws increase from 0.5 N to approximately 4 N. This indicates that as the climbing angle increases, the robot’s limbs tend to draw closer to the body’s centerline, creating a motion pattern similar to ‘hugging’. In the experimental results, however, the lateral forces did not vary significantly with the increase in climbing angle, remaining around 4–5 N. We attribute the higher lateral forces observed in low-angle experiments compared to the simulation results to the system errors of the robot itself and the friction between the chassis and the surface, which causes lateral disturbances, requiring greater force to maintain lateral stability. Both the simulation and experimental results indicate that the tailless condition generates greater lateral forces compared to the tailed condition.

For the tangential forces $F_{y}$, the simulation results indicate that as the climbing angle increases, the tangential forces at the feet steadily rise. The tangential force at the front claws increases from 4–5 N to 17–20 N, while the rear claws see an increase from 4–5 N to 9–12 N. The difference between the front and rear claws widens as the angle increases, indicating that the front claws bear a greater dragging force. The experimental results follow the same trend as the simulation but are slightly higher than the simulation values. This discrepancy is likely due to the robot’s chassis contacting the rough surface during actual tests, which requires the robot to overcome additional friction between the chassis and the surface. The tangential force component typically increases as the slope angle increases. However, the experimental results in Figure 13d show that the tangential force on the hind foot decreases between 60° and 90°. We believe this may be due to the moment generated by the shift in the robot’s center of gravity on steeper slopes, which causes the rear part of the robot’s chassis to come into contact with the surface. This contact interferes with the hind foot’s proper engagement with the wall, reducing the number of spines in contact and consequently decreasing the tangential force on the hind foot. Additionally, both the simulation and experimental results show that the robot’s feet typically generate greater tangential forces in the tailless condition.

For the normal forces $F_{z}$, the simulation results show that as the climbing angle increases, the normal forces at the feet gradually decrease. When the climbing angles are 0°, 30°, and 60°, the normal forces are positive, indicating that the feet are supported by the wall. However, at climbing angles of 86° and 90°, the normal forces become negative, indicating that the feet are relying on the adhesion of the spiny claws. In the tailless condition, the normal forces at the rear claws are greater than those at the front claws, due to the lack of tail support to distribute the load. The experimental results are similar to the simulation in both trend and magnitude, demonstrating the accuracy of the simulation results.

In addition to GRF testing, we also evaluated the impact of tail removal on the robot’s climbing speed. We controlled the robot’s speed by adjusting its step length using a remote controller and measured the time it took for the robot to cover a certain distance to calculate its actual speed. Five experiments were conducted at each slope, and the results are shown in Figure 14. With the tail, the robot achieved maximum speeds of 15.4 cm/s, 14.1 cm/s, 8.5 cm/s, and 2.12 cm/s on slopes of 0°, 30°, 60°, and 90°, respectively. After removing the tail, the robot’s maximum speeds at 0°, 30°, and 60° were 15.2 cm/s, 13.8 cm/s, and 7.9 cm/s, respectively, which are lower than the speeds with the tail. Additionally, the robot was unable to maintain stable climbing on a 90° vertical surface without the tail.

## 5. Discussion and Conclusions

Here we developed a gecko-inspired bionic climbing robot and constructed a corresponding climbing simulation system. This system allows us to set various motion parameters and inclination angles in a simulated environment, enabling the simulation of the robot’s climbing behavior with and without a tail, as well as predicting potential climbing failures. Additionally, we built an adjustable-angle force measurement platform to test the contact forces and movement speed of the physical prototype on different slopes. Through simulations and experimental tests, we observed differences in the robot’s performance with and without a tail. Based on the experimental results, we will discuss the impact of the tail on the robot’s locomotion ability in terms of balance, stability, and maneuverability.

Extensive research on gecko tails has shown that the tail plays a crucial role in maintaining balance during locomotion. When a gecko’s foot slips, its tail moves downward to maintain balance, preventing it from tipping backward and allowing it to continue climbing on vertical surfaces [17]. In our tests of the robot’s vertical climbing performance, we also validated the balancing role of the tail. Both the simulation and experimental results indicate that after the robot’s tail is removed, the body tends to pitch backward during climbing, causing the front claws to slip, which further leads to a loss of adhesion in the front claws, ultimately resulting in the robot falling from the wall. Furthermore, from the movement trajectory of the robot’s center of mass in the normal direction, it can be observed that after the tail is removed, the center of mass exhibits greater amplitude in the normal direction. During climbing, this normal displacement of the center of mass increases the pitch moment, significantly raising the risk of pitch failure. The forces experienced by the robot’s feet during climbing reflect the tail’s role in suppressing the pitch moment. Compared to the tailless condition, the robot with a tail exhibits a smaller difference in contact forces between the front and rear claws, meaning the front claws require less adhesive force to maintain the balance of the body.

Geckos can maintain their stability while moving quickly on horizontal and vertical surfaces by increasing the lateral undulation amplitude and frequency of their bodies and tails [19,20]. Although the climbing robot we designed does not have an active tail or flexible body, comparative experiments reveal that, after the tail is removed, the robot experiences greater slippage in the tangential direction and larger lateral oscillations during climbing. In the tailless condition, the robot’s increased lateral oscillations are also reflected in the contact forces at its claws. Both simulation and experimental results indicate that, in the absence of a tail, the robot generates greater lateral forces at the claws to maintain stable climbing. These larger lateral forces also make the robot more prone to lateral disturbances. Since we measured the robot’s average contact forces during stable attachment, and the robot uses the same gait with similar total displacement, larger GRFs at the foot-end throughout the climbing process result in greater work being done and, consequently, higher energy consumption. Compared to the tailless condition, the presence of a tail reduces the contact forces at the foot-end. Therefore, the tail can improve motion efficiency to some extent.

In nature, geckos sometimes detach their tails as a defense mechanism to distract predators [29,30]. However, studies on tailless geckos or lizards have found that losing the tail reduces their maneuverability and decreases their locomotion speed [31]. This aligns with our experimental findings. By comparing the climbing speed of the robot with and without a tail, we observed that the robot’s climbing speed decreases on different slopes after the tail is removed.

In this study, we utilized a dynamic climbing model and experimental platform to compare the robot’s performance with and without a tail through both simulations and experiments. The results indicate that the tail significantly enhances the robot’s performance in three key areas: balance, stability, and maneuverability. Specifically, the tail helps maintain balance during climbing, improves the robot’s stability on various slopes, and enhances its maneuverability. In future studies, we plan to further investigate the effects of tail length, mass distribution, and material composition on the climbing performance of the robot. This includes analyzing how different lengths and dimensions of the tail impact the robot’s balance, stability, and energy consumption. Specifically, tail length may affect the position of the center of mass, the magnitude of pitch moments, and the distribution of contact forces, which will influence the robot’s performance on various slopes and surface conditions. A soft tail material may effectively dissipate impact energy, reduce oscillations, and better adapt to irregular surfaces. By optimizing the geometric design of the tail, we aim to further enhance the climbing efficiency and adaptability of the robot.

## Figures and Tables

**Figure 1 biomimetics-09-00625-f001:**
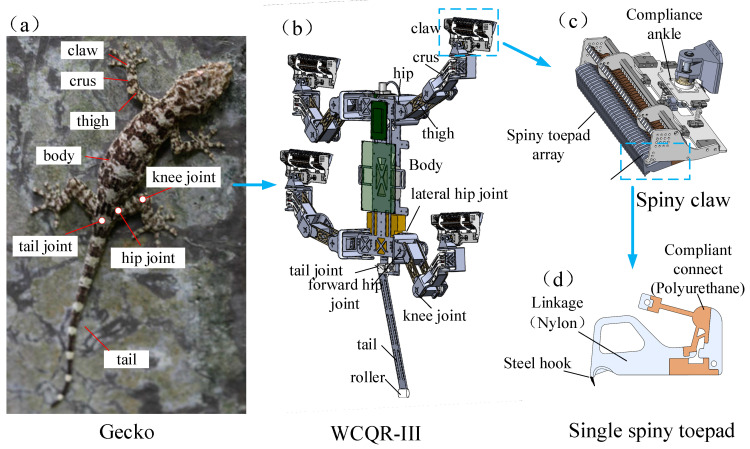
(**a**) Overall structure of the gecko, (**b**) Overall structure of the gecko-inspired robot WCQR-III, (**c**) The structure of the spiny claw, and (**d**) The structure of a single spiny toepad.

**Figure 2 biomimetics-09-00625-f002:**
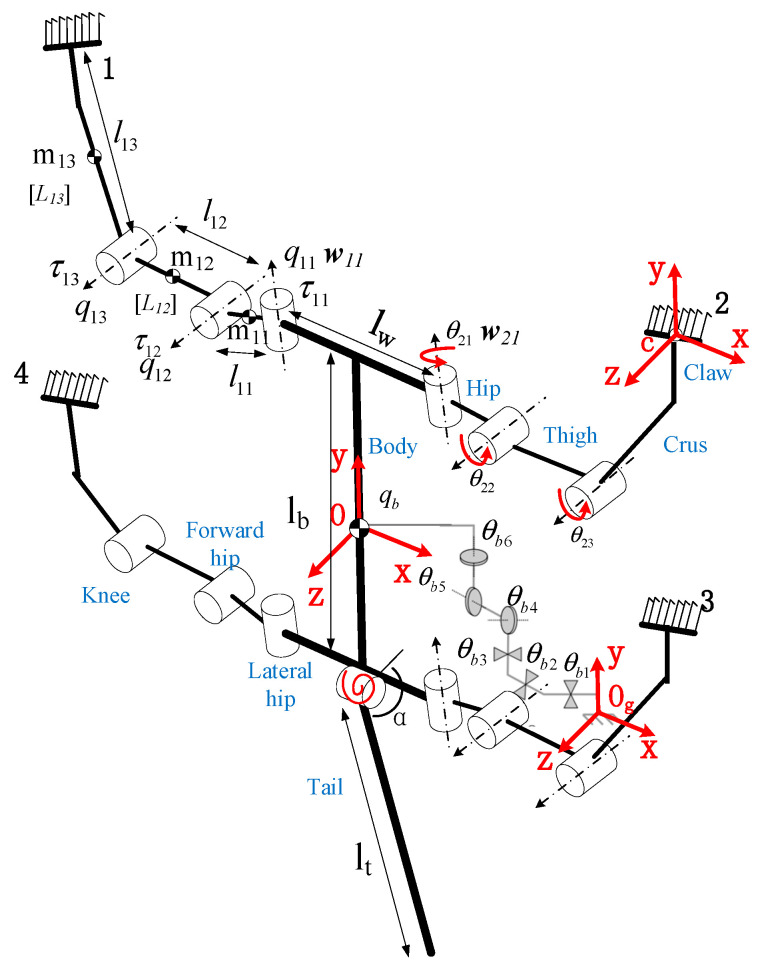
The topology of the WCQR-III robot.

**Figure 3 biomimetics-09-00625-f003:**
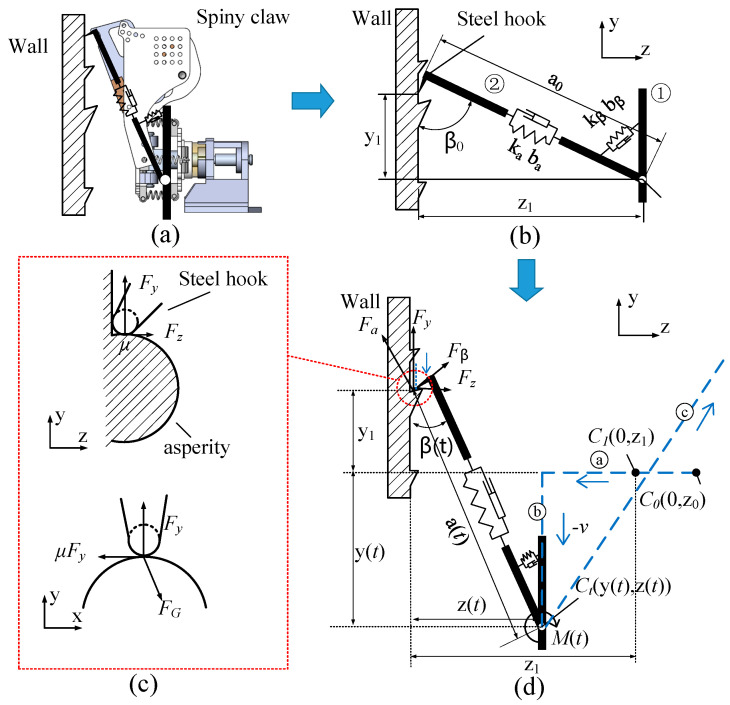
Dynamic contact model of spiny toepad: (**a**) the spiny claw; (**b**) hybrid spring-damper linkage model; (**c**) attachment process; (**d**) forces on the steel hook and rough surface.

**Figure 4 biomimetics-09-00625-f004:**
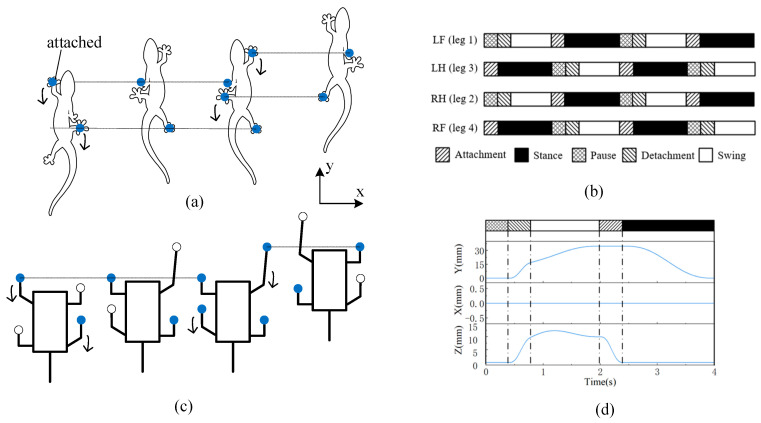
(**a**) Gecko’s diagonal gait; (**b**) Duty cycle of each phase within a gait cycle; (**c**) Robot’s climbing gait; (**d**) Foot coordinates during a gait cycle.

**Figure 5 biomimetics-09-00625-f005:**
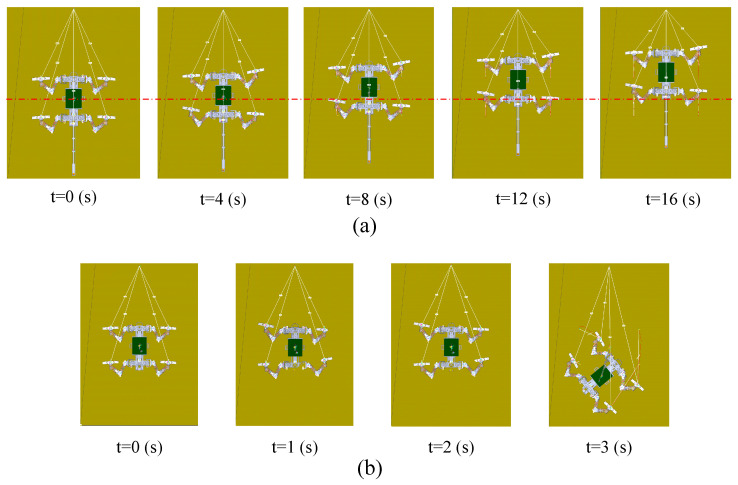
Simulation of the robot’s climbing test on a 90° vertical surface: (**a**) With tail; (**b**) Without tail.

**Figure 6 biomimetics-09-00625-f006:**
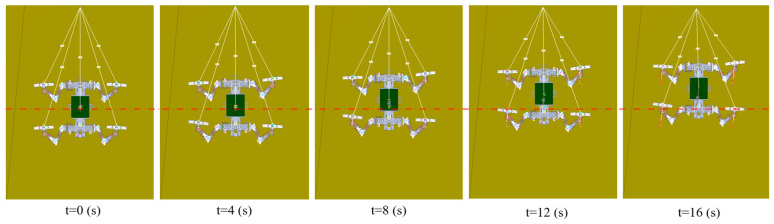
The robot stably climbing an 86° slope in the simulation environment.

**Figure 7 biomimetics-09-00625-f007:**
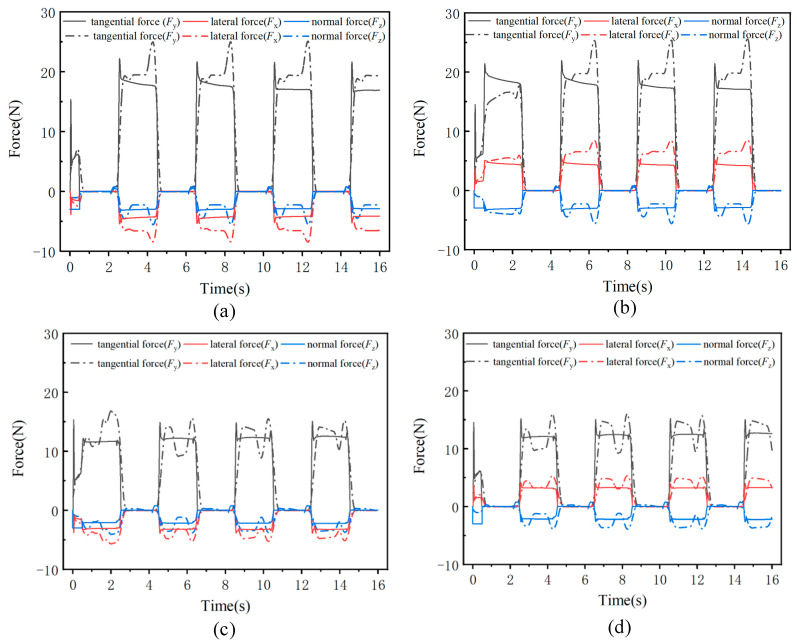
Comparison of foot-end contact forces during surface climbing with and without a tail: (**a**) Left front foot (**b**) Right front foot (**c**) Left hind foot (**d**) Right hind foot.

**Figure 8 biomimetics-09-00625-f008:**
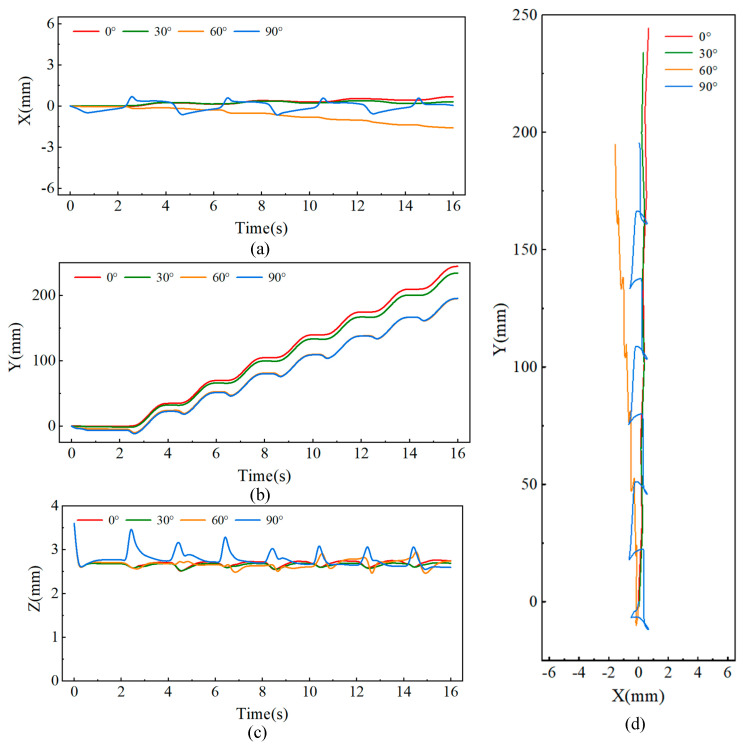
Center of mass displacement of the robot in the tailed condition while climbing at different angles: (**a**) Lateral displacement, (**b**) Vertical displacement, (**c**) Normal displacement, (**d**) CoM trajectory in the x-y plane.

**Figure 9 biomimetics-09-00625-f009:**
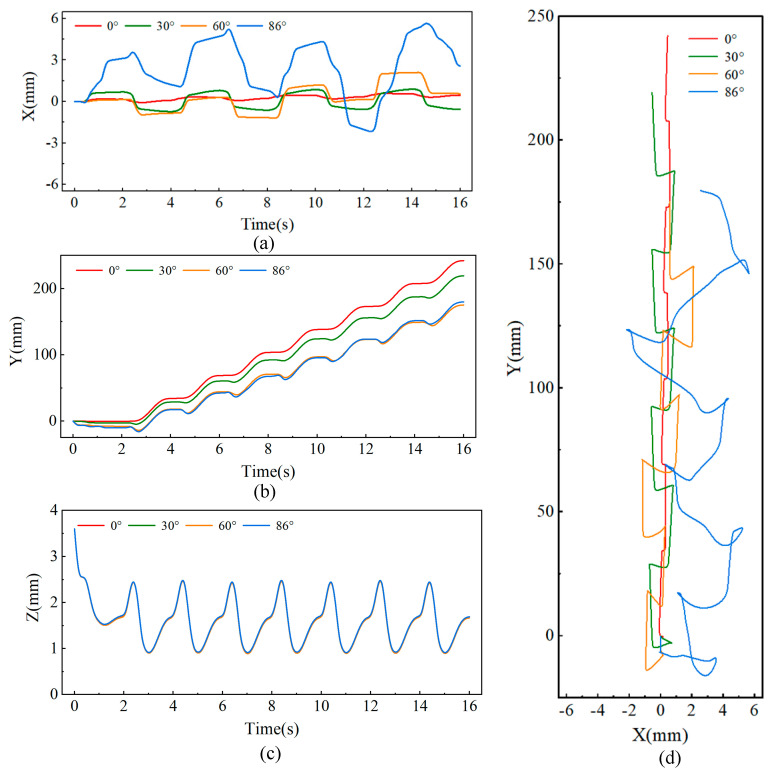
Center of mass displacement of the robot in the tailless condition while climbing at different angles: (**a**) Lateral displacement, (**b**) Vertical displacement, (**c**) Normal displacement, (**d**) CoM trajectory in the x-y plane.

**Figure 10 biomimetics-09-00625-f010:**
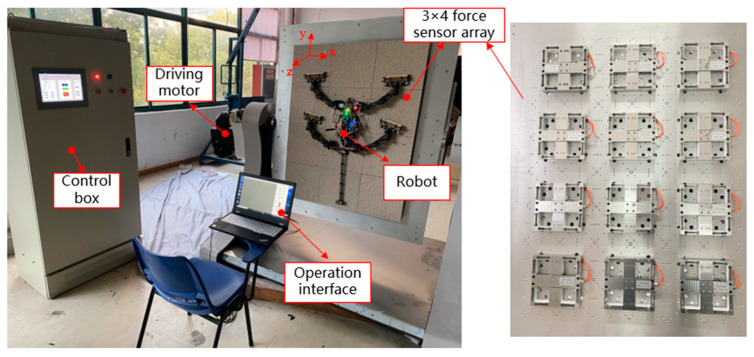
Adjustable angle contact force measurement platform.

**Figure 11 biomimetics-09-00625-f011:**
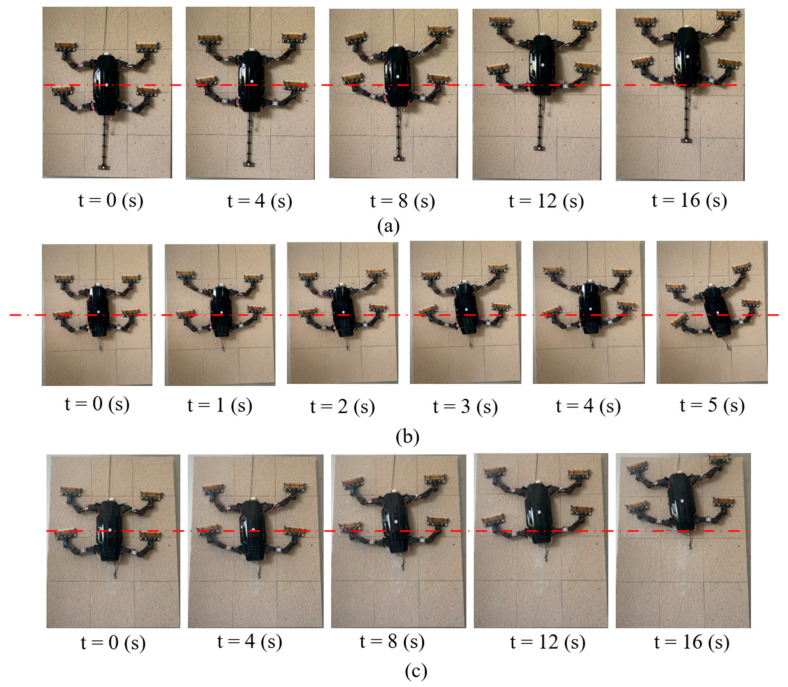
(**a**) Time sequence of the robot climbing a 90° surface with a tail; (**b**) Time sequence of the robot climbing a 90° surface without a tail; (**c**) Time sequence of the robot climbing a 82° surface without a tail.

**Figure 12 biomimetics-09-00625-f012:**
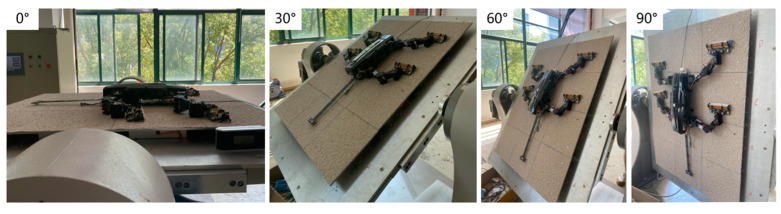
The robot climbing on rough surfaces with different incline angles.

**Figure 13 biomimetics-09-00625-f013:**
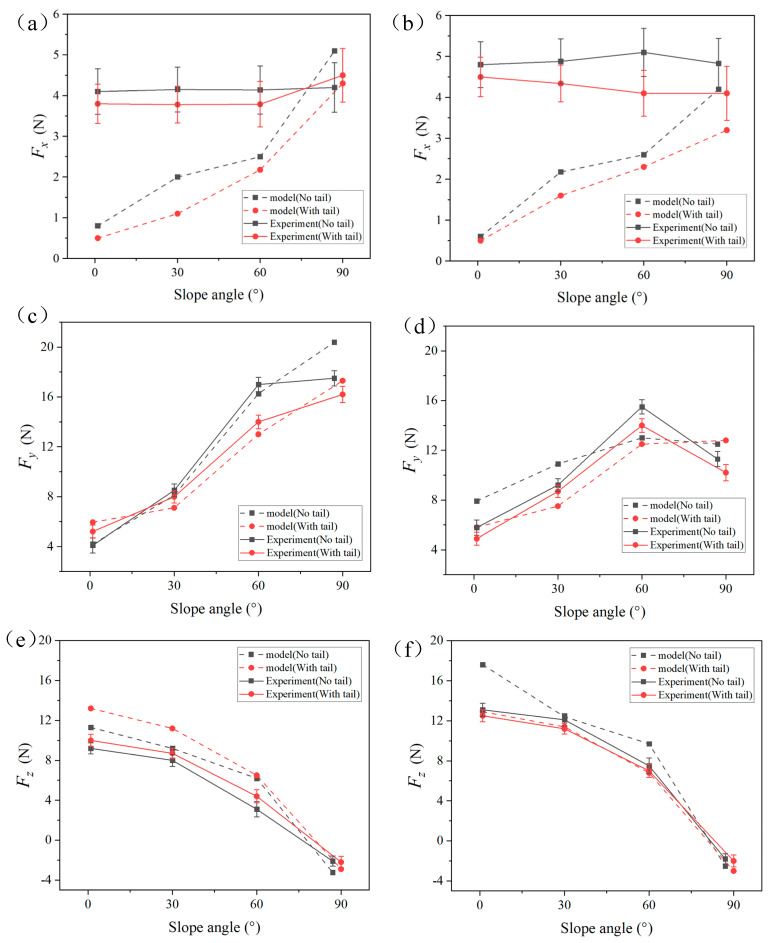
GRF generated by the robot at different angles with and without a tail: (**a**) Lateral force of the front claws, (**b**) Lateral force of the rear claws, (**c**) Tangential force of the front claws, (**d**) Tangential force of the rear claws, (**e**) Normal force of the front claws, (**f**) Normal force of the rear claws. Error bars show standard errors.

**Figure 14 biomimetics-09-00625-f014:**
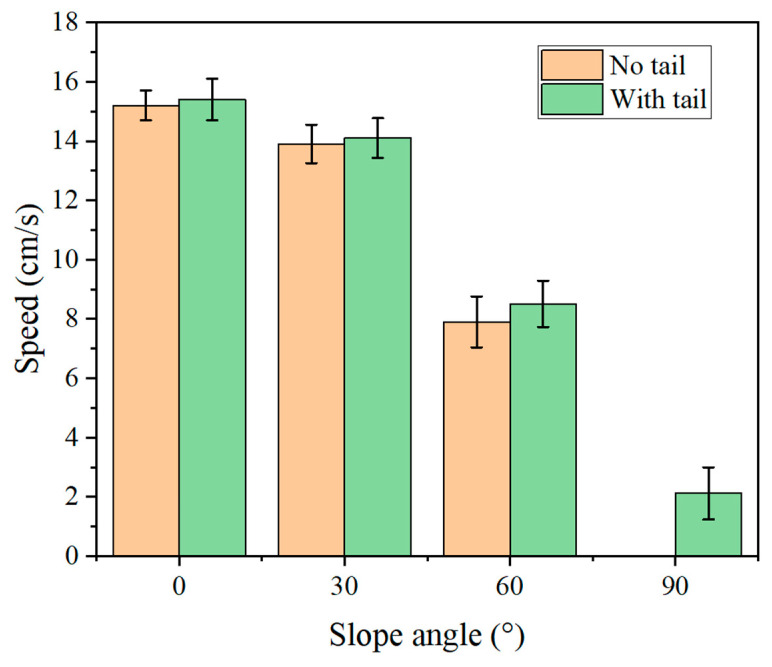
Maximum Speed of the Robot at Different Angles. Error bars show standard errors.

**Table 1 biomimetics-09-00625-t001:** Physical parameters of the robot.

Part	Mass (g)	Length (mm)
Body	2200	288 × 137
Hip	40.1	53
Thigh	203.5	100
Crus	72.9	100
Claw	231.1	100
Tail	20.3	300

**Table 2 biomimetics-09-00625-t002:** Parameters of spiny toepad.

Parameter	Value	Parameter	Value
$k_{a}$	191 N/m	$k_{\beta }$	0.0015 Nm/radian
$b_{a}$	0.42 Ns/m	$b_{\beta }$	0.02 Ns/m
$a_{0}$	30 mm	$\beta_{0}$	30°
μ	0.25	*P*	20%
$\beta_{app}$	90°	$\beta_{detach}$	30°

**Table 3 biomimetics-09-00625-t003:** Parameters of the measuring platform.

Number of Sensors	3 × 4
Sensors Models	DYDW-005
Measurement Range	±100 N
Individual Sensor Size	20 cm × 20 cm
Measurement Accuracy	0.2%F.S
Sampling Frequency	90 Hz
Rotation Angle	0~90°

## Data Availability

The raw data supporting the conclusions of this article will be made available by the authors on request.

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
