# Peer review of "Role of Tail Dynamics on the Climbing Performance of Gecko-Inspired Robots: A Simulation and Experimental Study"

_biomimetics, 2024, doi:10.3390/biomimetics9100625_

Round 1

Reviewer 1 Report

Comments and Suggestions for Authors

The paper is well-written and structured.

In the following some points to be addressed:

Add to Fig. 8 an indication of what is shown in dashed and continuous lines, as done in the text

Add to Fig. 14 an indication of what is shown in dashed and continuous lines, as done in the text

Please clarify what is shown in Fig 14, by explaining in the text and providing a picture showing the force trends at a certain angle (as an instance at 30°) for both configurations. I guess that the authors measured GRFs during the movement, thus acquiring the time histories from the sensors, but for each angle, the provided result is an average with an error band. Without the requested clarifications, the reader cannot understand properly the results in Fig. 14.

Please add a figure showing the time history of the lateral forces Fx , to support the assumption provided in lines 436-439, to justify the discrepancy between the modelling and the experimental result.

The sentence in lines 449-450 should be rephrased as “Additionally, both the simulation and experimental results show that the robot's feet generate in general, greater tangential forces in the tailless condition but in some conditions, measured values are compatible. This is because measurement results are overlapping in Figure 14.

How can the deviation in Fig. 14 d between the model (tailless condition) and the experimental result be explained? Please clarify and add an explanation to the discussion.

Please add a comment in the text about Figure 15 which shows that considering the variability of the measured speed no significant advantage is obtained in terms of climbing speed, given that the results shown in the figure provide compatible values between the two tested conditions.

In the discussion, the authors claim that the presence of a tail can lead to an improvement in efficiency (lines 508-510), based on the measured forces of Fig. 14. The conclusion is reasonable, but should be supported by quantifying the energy consumption (based on the measured time history of the forces) of the present study.

Moreover, how much the obtained result can be extrapolated to different operative scenarios (ground type, humidity)? Please add a comment in the text.

Comments on the Quality of English Language

No comments.

Author Response

Comments 1: Add to Fig. 8 an indication of what is shown in dashed and continuous lines, as done in the text.

Response 1:  Thank you for your suggestion. We have revised Fig. 8 according to your feedback and added an explanation in the figure caption to clarify what the solid and dashed lines represent.

 Fig.8 Comparison of foot-end contact forces during surface climbing with and without a tail: (a) Left front foot (b) Right front foot (c) Left hind foot (d) Right hind foot

Comments 2: Add to Fig. 14 an indication of what is shown in dashed and continuous lines, as done in the text.

Response 2: Thank you for your suggestion. We have revised Fig. 14 to include a clear indication of what the dashed and continuous lines represent, as described in the text.

Fig.14 GRF generated by the robot at different angles with and without a tail: (a) Lateral force of the front claws, (b) Lateral force of the rear claws, (c) Tangential force of the front claws, (d) Tangential force of the rear claws, (e) Normal force of the front claws, (f) Normal force of the rear claws. Error bars show standard errors.

Comments 3: Please clarify what is shown in Fig 14, by explaining in the text and providing a picture showing the force trends at a certain angle (as an instance at 30°) for both configurations. I guess that the authors measured GRFs during the movement, thus acquiring the time histories from the sensors, but for each angle, the provided result is an average with an error band. Without the requested clarifications, the reader cannot understand properly the results in Fig. 14.

Response 3: Thank you for your suggestion. We have provided a more detailed explanation of the experimental measurements in Figure 14. Since the robot uses a diagonal gait, the contact forces on the left and right feet are the same in magnitude but offset by one cycle. The figure below shows the foot-end contact forces of the left front and right hind feet over a 16-second climbing period when the robot climbs vertically at 90°, with the shaded areas indicating the error bands.

In the experiment, we measured the ground reaction forces (GRFs) in three directions after the robot achieved stable attachment, conducting five measurements in total. Similarly, we also tested the GRFs after stable attachment when the robot climbed at different angles.

For ease of comparison, we did not present the time-varying GRF curves for each angle. Instead, Figure 14 displays the average GRFs and the corresponding error ranges for each angle.

In the revised manuscript, we further clarified the explanation of Figure 14:

"Since the robot uses a diagonal gait, the contact forces on the left and right feet are the same in magnitude but offset by one cycle. After the robot achieved stable attachment, we measured the GRFs at different angles, conducting five measurements in total. All data are presented as average values with error bands."

Fig. 1 Ground reaction forces (GRFs) of the robot's foot-end on a 90° rough surface: (a) Ground reaction forces of the left front foot; (b) Ground reaction forces of the right hind foot.

Comments 4: Please add a figure showing the time history of the lateral forces Fx , to support the assumption provided in lines 436-439, to justify the discrepancy between the modelling and the experimental result.

Response 4:  Thank you for your suggestion. We measured the ground reaction forces (GRFs) after the robot achieved stable attachment while climbing at different angles and recorded the relevant data. As shown in Figure 14, the lateral force Fx measured at lower angles is greater than the simulation results. We believe this discrepancy mainly arises because the simulation did not account for the friction between the robot's body and the wall. In the experiment, friction between the robot's chassis and the rough surface introduced additional lateral interference, requiring a greater lateral force to maintain stability. Therefore, we believe that Figure 14 sufficiently addresses the related issue.

Comments 5: The sentence in lines 449-450 should be rephrased as “Additionally, both the simulation and experimental results show that the robot's feet generate in general, greater tangential forces in the tailless condition but in some conditions, measured values are compatible. This is because measurement results are overlapping in Figure 14.

Response 5: Thank you for your suggestion. The overlapping of error bars is due to measurement errors. However, the average results indicate that, in the tailless condition, the robot's feet typically generate greater tangential forces. We have revised the sentence in the manuscript as follows:

“Additionally, both the simulation and experimental results show that the robot's feet typically generate greater tangential forces in the tailless condition.”

Comments 6: How can the deviation in Fig. 14 d between the model (tailless condition) and the experimental result be explained? Please clarify and add an explanation to the discussion.

Response 6: Thank you for your suggestion. In the revised manuscript, we have added an explanation for the deviation between the simulation and experimental results in Figure 14. The tangential force component typically increases as the slope angle increases. However, the experimental results in Figure 14d show that the tangential force on the hind foot decreases between 60° and 90°. We believe this may be due to the moment generated by the shift in the robot's center of gravity on steeper slopes, which causes the rear part of the robot's chassis to come into contact with the surface. This contact interferes with the hind foot’s proper engagement with the wall, reducing the number of spines in contact and consequently decreasing the tangential force on the hind foot.

Comments 7: Please add a comment in the text about Figure 15 which shows that considering the variability of the measured speed no significant advantage is obtained in terms of climbing speed, given that the results shown in the figure provide compatible values between the two tested conditions.

Response 7: Thank you for your comment. The robot's speed was calculated by measuring the time it took to cover a certain distance. Due to the acceleration phase as the robot moves from a stationary position, and the short travel distance imposed by the test platform, there is a noticeable variation in the measured data, resulting in relatively large error bars. However, the experimental results show that the average speed with the tail is greater than without the tail in five trials. Furthermore, as the slope angle increases, the speed difference becomes more pronounced.

At shallower slopes, the effect of the tail on speed is minimal. However, as the slope steepens, the speed in the tailless condition becomes noticeably lower compared to the tailed condition. This is because the tail mainly affects the robot's stability in the pitch degree of freedom. On flatter slopes, deviations in pitch have little impact on stability. But as the slope becomes steeper, any slight variation in the pitch angle significantly affects climbing stability. When the slope reaches 90 degrees (vertical), the tailless condition leads to substantial pitch deviations during climbing, causing the center of mass to rise and leading to pitch failure.

Therefore, we believe it is important to retain the conclusion that the tail improves the robot's climbing speed, particularly on steeper slopes.

Comments 8: In the discussion, the authors claim that the presence of a tail can lead to an improvement in efficiency (lines 508-510), based on the measured forces of Fig. 14. The conclusion is reasonable, but should be supported by quantifying the energy consumption (based on the measured time history of the forces) of the present study.

Response 8: Thank you for your suggestion. Since we measured the robot's average contact forces during stable attachment, and the robot uses the same gait with similar total displacement, larger ground reaction forces (GRFs) at the foot-end throughout the climbing process indicate that more work is required, leading to higher energy consumption. We have added the following explanation in the revised manuscript:

"Since we measured the robot's average contact forces during stable attachment, and the robot uses the same gait with similar total displacement, larger GRFs at the foot-end throughout the climbing process result in greater work being done and, consequently, higher energy consumption. Compared to the tailless condition, the presence of a tail reduces the contact forces at the foot-end. Therefore, the tail can improve motion efficiency to some extent."

Comments 9: Moreover, how much the obtained result can be extrapolated to different operative scenarios (ground type, humidity)? Please add a comment in the text.

Response 9: Thank you for your insightful comment. The attachment mechanism used by the robot in this study is based on microspines, which are designed to adapt to rough surface environments and are less affected by dust and moisture [x]. However, the effectiveness of this mechanism depends on the roughness of the surface, which is influenced by the tip diameter of the steel hooks at the end of the microspines [x]. Stable adhesion can only be achieved when the diameter of the rough surface particles exceeds the tip diameter of the steel hooks. In general, the larger the surface roughness, the higher the probability of forming stable adhesion. The microspines used in this study have steel hooks with a tip diameter of 100 μm, making them suitable for climbing surfaces with roughness greater than 100 μm.

[1]     Z. Dai, S. N. Gorb, and U. Schwarz, "Roughness-dependent friction force of the tarsal claw system in the beetle Pachnoda marginata (Coleoptera, Scarabaeidae)," Journal of Experimental Biology, vol. 205, no. 16, pp. 2479-2488, 2002.

[2]     A. T. Asbeck, S. Kim, M. R. Cutkosky, W. R. Provancher, and M. Lanzetta, "Scaling hard vertical surfaces with compliant microspine arrays," The International Journal of Robotics Research, vol. 25, no. 12, pp. 1165-1179, 2006.

Reviewer 2 Report

Comments and Suggestions for Authors

1. It would be better to have a pseudocode of Fig.4 Flowchart of the Dynamic Climbing Algorithm

2. The tail length and dimensions have not been studied in the paper. 

3. What if the tail is made of soft material? How this affect the performance? What are the challenges? 

Author Response

Comments 1: It would be better to have a pseudocode of Fig.4 Flowchart of the Dynamic Climbing Algorithm.

Response 1: Thank you for your valuable feedback on our manuscript. You suggested providing a pseudocode for Fig.4, "Flowchart of the Dynamic Climbing Algorithm," and we completely agree with your suggestion. Accordingly, we have added the pseudocode for Fig.4 in the revised manuscript to help readers better understand the workings of the algorithm.

 Fig.4 Dynamic Climbing Algorithm

Comments 2: The tail length and dimensions have not been studied in the paper.

Response 2: Thank you for your valuable feedback. You pointed out that we have not studied the tail length and dimensions in detail in the paper. We agree that this is an important research direction worth exploring further. Indeed, the length and geometric dimensions of the tail may significantly impact the balance, stability, and maneuverability of the climbing robot.

In this study, our primary objective was to investigate the effect of the presence or absence of a tail on the robot's performance, and therefore, we did not analyze the specific geometric parameters of the tail in detail. However, studies have shown that after a gecko loses its tail, its climbing speed increases as the tail regrows, indicating a significant correlation between the tail length and its functionality [1]. In future research, we plan to further study the influence of tail length, mass distribution, and other geometric characteristics on the robot's locomotion ability, through both experimental and simulation-based quantitative analysis.

We have added the following content to the revised manuscript in the “5. Discussion and Conclusion” section:

“In future studies, we plan to further investigate the effects of tail length, mass distribution, and material composition on the climbing performance of the robot. This includes analyzing how different lengths and dimensions of the tail impact the robot’s balance, stability, and energy consumption. Specifically, the tail length may affect the position of the center of mass, the magnitude of pitch moments, and the distribution of contact forces, which will influence the robot's performance on various slopes and surface conditions. A soft tail material may effectively dissipate impact energy, reduce oscillations, and better adapt to irregular surfaces. By optimizing the geometric design of the tail, we aim to further enhance the climbing efficiency and adaptability of the robot.”

[1]     K. Jagnandan, A. P. Russell, and T. E. Higham, "Tail autotomy and subsequent regeneration alter the mechanics of locomotion in lizards," Journal of Experimental Biology, vol. 217, no. 21, pp. 3891-3897, 2014.

Comments 3: What if the tail is made of soft material? How this affect the performance? What are the challenges?

Response 3: Thank you for your valuable suggestion. We believe that a soft material tail does indeed hold potential for climbing robots. A soft tail could effectively dissipate impact energy, reduce oscillations, and better adapt to irregular surfaces. However, due to its high deformability, controlling the motion and shape of a soft tail may become more complex, likely requiring more precise sensors and algorithms for real-time adjustment and feedback control.

In the literature [1], previous studies have reviewed and summarized the different impacts of soft and rigid tails on the performance of climbing robots. A rigid tail can provide greater preload and stability by adjusting its angle to shift the center of mass, enhancing stability during climbing. In contrast, a soft tail, with its greater adaptability and flexibility, can increase the contact area on irregular surfaces, improving adhesion. However, a soft tail may also be more prone to wear and tear. Therefore, future research could explore the design of a soft material tail for climbing robots and compare the performance of rigid versus soft tails through simulations and experiments to validate the role of a soft tail in climbing performance.

We have added the following content to the revised manuscript in the “5. Discussion and Conclusion” section:

“In future studies, we plan to further investigate the effects of tail length, mass distribution, and material composition on the climbing performance of the robot. This includes analyzing how different lengths and dimensions of the tail impact the robot’s balance, stability, and energy consumption. Specifically, the tail length may affect the position of the center of mass, the magnitude of pitch moments, and the distribution of contact forces, which will influence the robot's performance on various slopes and surface conditions. A soft tail material may effectively dissipate impact energy, reduce oscillations, and better adapt to irregular surfaces. By optimizing the geometric design of the tail, we aim to further enhance the climbing efficiency and adaptability of the robot.”

[1]  G. Zang, Z. Dai, and P. Manoonpong, "The roles and comparison of rigid and soft tails in gecko-inspired climbing robots: A mini-review," Frontiers in Bioengineering and Biotechnology, vol. 10, p. 900389, 2022.
